# Using XGBoost Regression to Analyze the Importance of Input Features Applied to an Artificial Intelligence Model for the Biomass Gasification System

Hung-Ta Wen [1], Hom-Yu Wu [2] and Kuo-Chien Liao [1,*]

1 Department of Aeronautical Engineering, Chaoyang University of Technology, Taichung 413, Taiwan
2 Department of Mechanical Engineering, Lungwa University of Science and Technology, Taoyuan 333, Taiwan
* Correspondence: james19831111@gmail.com; Tel.: +886-982-365-503

**Abstract:** Recently, artificial intelligence models have been developed to simulate the biomass gasification systems. The extant research models use different input features, such as carbon, hydrogen, nitrogen, sulfur, oxygen, and moisture content, in addition to ash, reaction temperature, volatile matter (VM), a lower heating value (LHV), and equivalence ratio (ER). The importance of these input features applied to artificial intelligence models are analyzed in this study; further, the XGBoost regression model was used to simulate a biomass gasification system and investigate its performance. The top-four features, according to the results are ER, VM, LHV, and carbon content. The coefficient of determination ($R^2$) was highest (0.96) when all eleven input features noted above were selected. Further, the model performance using the top-three features produced a $R^2$ value of 0.93. Thus, the XGBoost model performance was validated again and observed to outperform those of previous studies with a lower mean-squared error of 1.55. The comparison error for the hydrogen gas composition produced from the gasification at a temperature of 900 °C and ER = 0.4 was 0.07%.

**Keywords:** artificial intelligence model; biomass gasification; lower heating value; equivalence ratio; feature importance analysis; XGBoost regression

## 1. Introduction

Alam and Qiao presented a review on the management of municipal solid waste (MSW) in Bangladesh; this study on the energy recovery from MSW showed that the wastes could be minimized by employing an appropriate MSW management system. According to their results, this project was estimated to effectively reduce the disposal costs by approximately USD 15.29 million annually [1]. Furthermore, Malkow proposed several approaches for pyrolysis and gasification technologies that could improve the fuel utilization and combustion efficiency in Europe [2]. Basu explored the design, analysis, and operational aspects of the biomass gasification from the perspective of thermochemical conversion [3]. Baruah and Baruah reported a review of the optimum parameters for a gasifier design that could achieve the best model performance during gasification [4]. Fixed-bed gasifiers are the simplest type, and the updraft-type gasifier is shown in Figure 1.

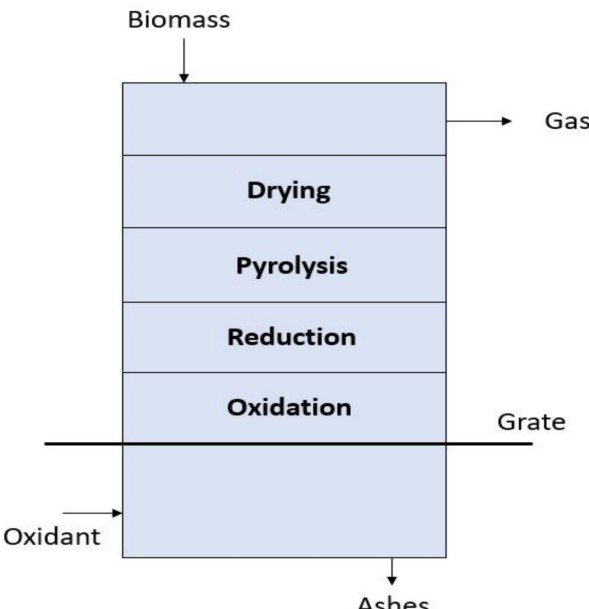

**Figure 1.** Fixed-bed updraft gasifier.

Recently, machine learning has been favored and used widely as an approach to model complex nonlinear systems. Ullah et al. used gene expression programming (GEP) to develop a model for the mix design relation of lightweight foamed concrete (LWFC) whose $R^2$ value for performance reached 0.95 [5]. Machine learning technologies, such as the random forest regression and GEP have also been used to simulate the depth of wear of ecofriendly concrete [6]. Furthermore, a support vector machine (SVM) and random forest have been employed to model predictions regarding the compressed LWFC [7]. A combination of the artificial neural network (ANN), SVM, and GEP has been used to model the mechanical properties of self-compacting concrete [8]. Bagasse-ash-based geopolymers have been investigated, along with different mixture rates of propylene fibers [9]. The predictions of individual and ensemble approaches for the compressive strength of fly-ash-based concrete have also been proposed [10]. Such models have been compared and optimized using individually learned and ensemble learned machine intelligence algorithms [11].

Puig-Arnavat et al. developed an ANN to model the biomass gasification process in a fluidized bed reactor and obtained successful predictions for the main producer gas during the complex chemical reactions [12]. Souza et al. presented a method based on ANNs to obtain the regression calculations for different kinds of biomass given the operating conditions; the maximum amounts of produced gas were investigated under different operating conditions [13].

ANNs and gradient boosting regression models were developed by Wen et al. to predict rice husk syngas compositions, using the equivalence ratio (ER), bottom temperature, and steam flow rate as the model input features [14]. An ANN model was developed by Baruah and Hazarika for the biomass gasification using six input features, namely the moisture content (MC), ash, C, H, O, and reaction temperature (Tg); the $R^2$ value for the model performance for the $H_2$ production was 0.9855 [15]. Furthermore, Ozonoh et al. proposed estimating the gasification efficiency by selecting two kinds of input feature numbers for the ANN models; the first model involved eleven features, namely the ash content, MC, volatile matter (VM), C, H, N, O, and S, in addition to a lower heating value (LHV), ER, and Tg; the second model utilized three features: C, VM, and Tg. The $R^2$ value for the performances of both models for the $H_2$ production was 0.95 [16].

Wen et al. proposed the feature importance analysis for $NO_x$ and $CO_2$ artificial intelligence models of diesel vehicles, to investigate the model accuracy, based on the different numbers of features. Finally, the best model and best prediction model were

determined in terms of the model performance [17]. In addition, the XGboost algorithm has been widely utilized in the detection, disease diagnosis, classification, and prediction [18–22].

The XGboost algorithm has been previously shown to have an excellent performance, which has also been approved and validated in other studies. The model performance is often affected when using different numbers of input features. This study aims to investigate the importance of the input features applied to an artificial intelligence model for the biomass gasification; further, the development of a XGBoost regression model for the biomass gasification system is presented.

## 2. Materials and Methods

### 2.1. Data

The data were collected from the extant research on biomass, coal, and the blend of coal and biomass [23–52]. The total number of samples obtained was 315. The proximate analysis of the data included the ash content, MC, and VM, and the ultimate analysis of the data included eight additional variables, namely the C, H, N, O, and S content, as well as LHV, ER, and Tg. The statistical analysis results between the model input features and target are visually shown in Figure 2. For instance, Figure 2a shows the univariate and bivariate distributions of the independent variable (N) versus the dependent variable ($H_2$). It is helpful to understand the dataset relationships by plotting the univariate and bivariate distributions, simultaneously.

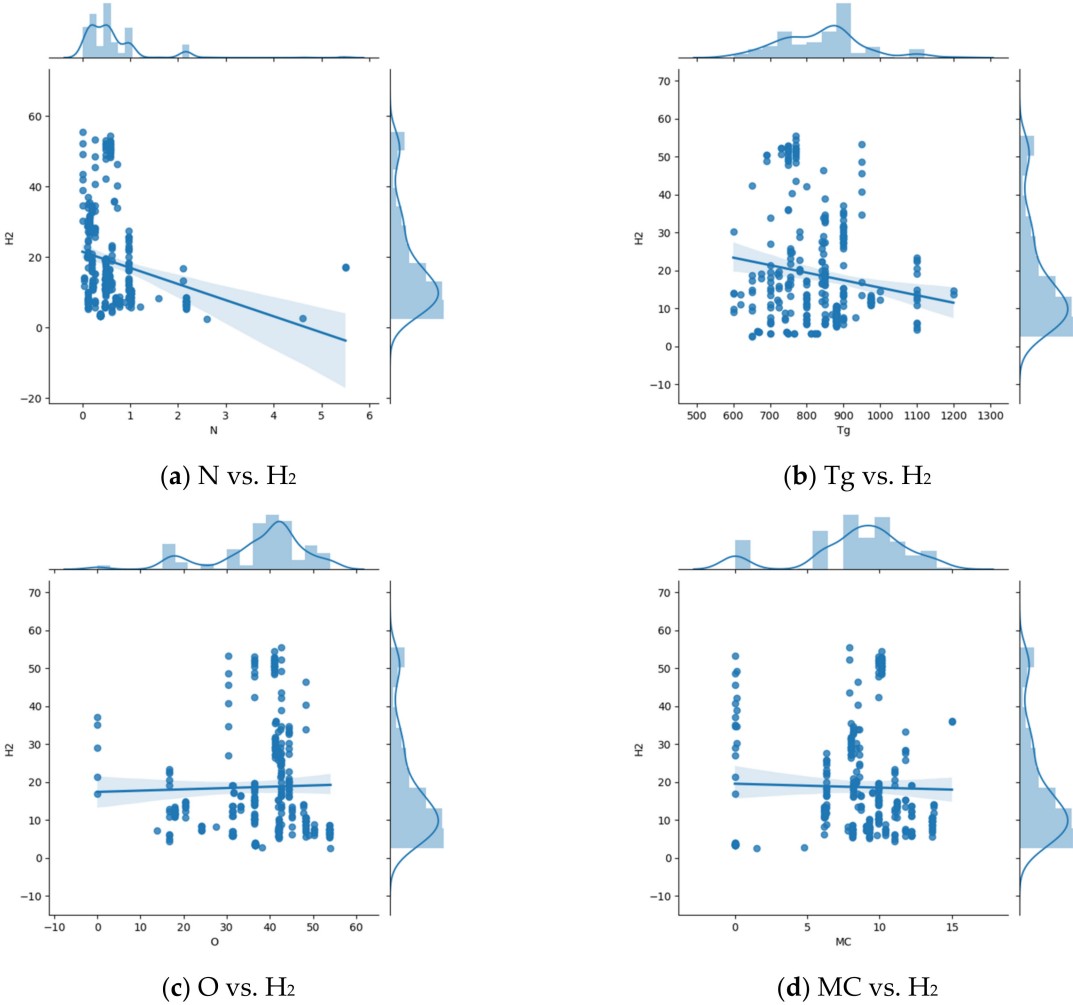

(**a**) N vs. $H_2$

(**b**) Tg vs. $H_2$

(**c**) O vs. $H_2$

(**d**) MC vs. $H_2$

**Figure 2.** *Cont.*

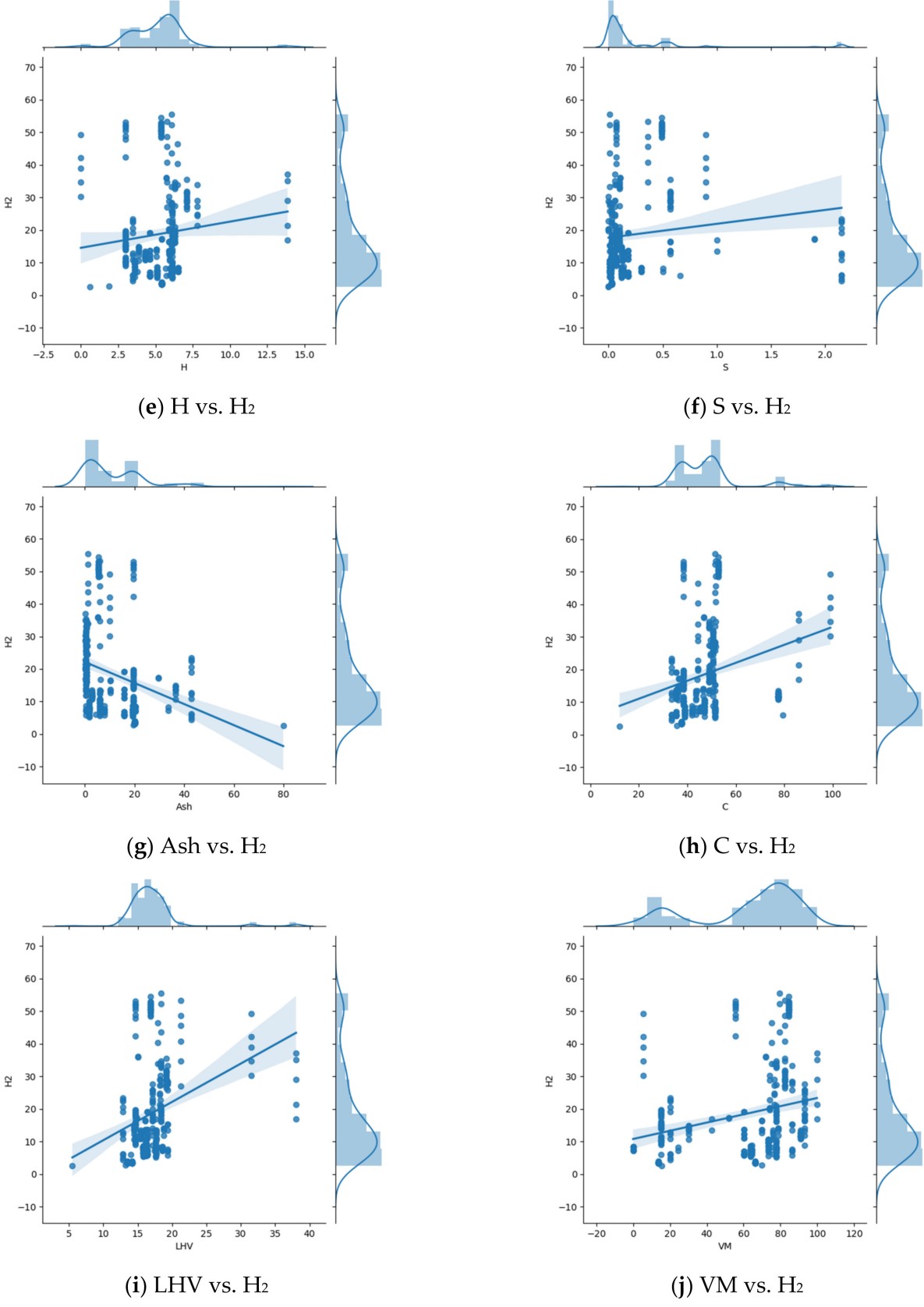

(**e**) H vs. H₂

(**f**) S vs. H₂

(**g**) Ash vs. H₂

(**h**) C vs. H₂

(**i**) LHV vs. H₂

(**j**) VM vs. H₂

**Figure 2.** *Cont.*

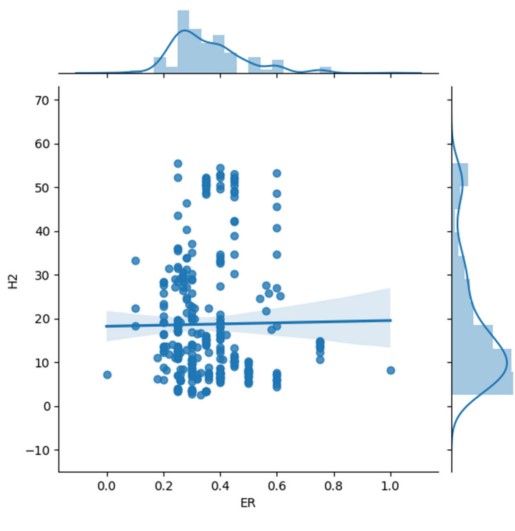

**(k)** ER vs. $H_2$

**Figure 2.** Statistical analyses between the model input features and the target.

### 2.2. Feature Importance

The permutation technique was used to analyze the rank and score of the feature importance. This analysis was implemented in Python, using permutation_importance directly. The process involved breaking the input feature values by random shuffling. The results showed the relative predictive power of these features in the model [53]. The algorithm for this procedure is as follows:

- Inputs: fitted predictive model $m$, tabular dataset (training or validation) $D$;
- Compute the reference score $s$ of the model $m$ on data $D$ (for instance, the accuracy for a classifier and $R^2$ for regression);
- For each feature $j$ (column of $D$);
- For each repetition $k$ in $1, \ldots \ldots, K$;
- Randomly shuffle column $j$ of data $D$ to generate a corrupted version of the data as $\widetilde{D}_{kj}$;
- Compute the score $s_{kj}$ of model $m$ on corrupted data $\widetilde{D}_{kj}$;
- Compute the importance $i_j$ for feature $f_j$ defined as

$$i_j = s - \frac{1}{K} \sum\nolimits_{k=1}^{K} s_{kj}$$

### 2.3. XGBoost Model

Chen and Guestrin proposed a novel sparsity-aware algorithm called the XGBoost, which is available as an open-source package for approximate tree learning. The XGBoost algorithm has been shown to reduce the calculation costs and provide a high model performance [54]. This algorithm has been widely used in many fields [55–57]. In this study, the data are split into two subsets using the Python sklearn library code, such that 70% is used as the training set, 10% is used for the validation, and 20% is used for the testing. Therefore, the XGBoost regression model can be trained using the training set, evaluated using the validation set, and then applied to the test set for the verification for the model performance evaluation.

When building the XGBoost regression model, several parameters should be assigned in the Python code. The main parameters of the XGBoost model are the following: Max_depth denoting the maximum depth of the tree, whose default value is 3; this was set to 15 in this study. N_estimators denote the number of trees used for boosting, whose value was set to 100 in the model, in this study. Learning_rate indicates the learning rate that

determines the step size at each iteration while moving toward a minimum loss function, and its value was set at 0.2 in this study. Colsample_bytree denotes a family of parameters for subsampling the columns within a range of 0 to 1, and this value was set at 0.3. Figure 3 shows the flowchart of the proposed method for the composition gas prediction during gasification.

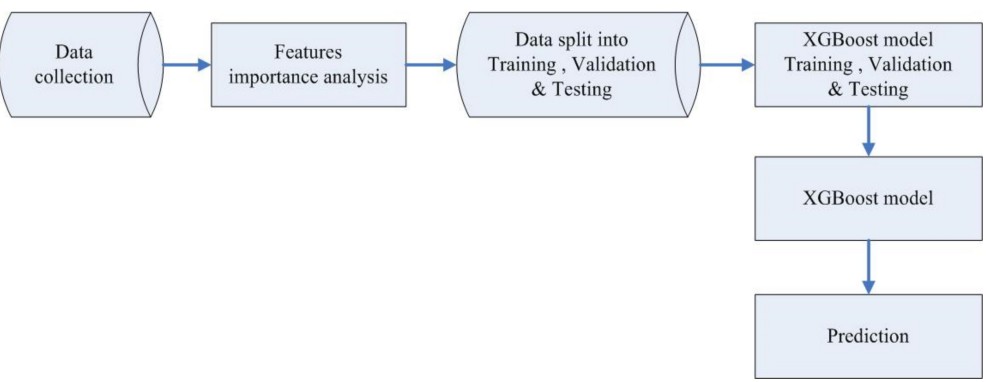

**Figure 3.** Flowchart of the proposed method for composition gas prediction in gasification.

Figure 4 shows the violin plots depicting the summary statistics and densities of C, VM, LHV, and $H_2$. The broader areas of the violin plots represent higher probabilities that the members of the population will take on the given values; correspondingly, the narrower areas represent the lower probabilities.

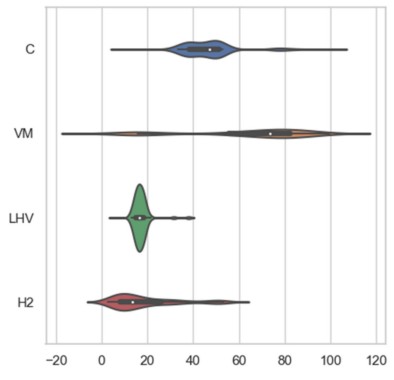
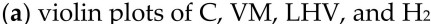

(**a**) violin plots of C, VM, LHV, and $H_2$

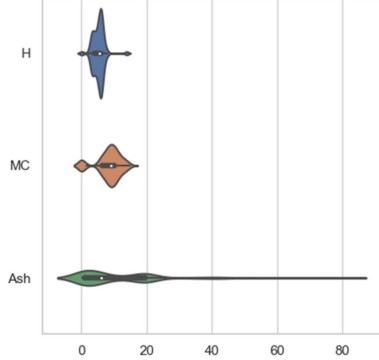

(**b**) violin plots of H, MC, and Ash

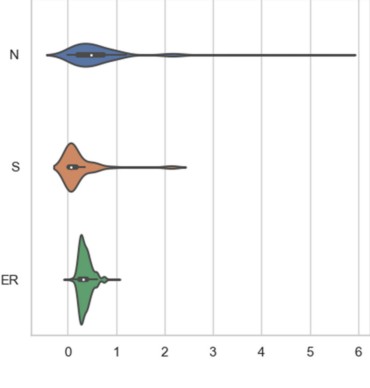

(**c**) violin plots of N, S, and ER

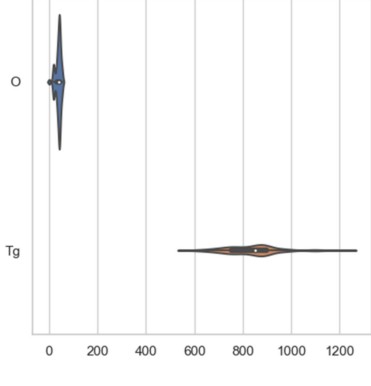

(**d**) violin plots of O and Tg

**Figure 4.** Violin plots of the model input features and target $H_2$.

The model performance reflects the accuracy of the model, and the mean absolute error (*MAE*), root mean-squared error (*RMSE*), and the coefficient of determination ($R^2$) are the metrics used in this study to assess the performance. The statistical equations for these metrics are given in Equations (1)–(3), where $P_i$ is the predicted value obtained with the model and $T_i$ is the measured value from the *PEMS*. $\overline{P}_i$ is the average of the predicted values for the entire dataset.

$$MAE = \frac{1}{n} \sum_{i=1}^{n} |T_i - P_i| \tag{1}$$

$$RMSE = \sqrt{\frac{1}{n} \sum_{i=1}^{n} (T_i - P_i)^2} \tag{2}$$

$$R^2 = 1 - \frac{\sum_{i=1}^{n} (P_i - T_i)^2}{\sum_{i=1}^{n} (\overline{P}_i - T_i)^2} \tag{3}$$

## 3. Results

### 3.1. Features Importance Analysis

The detailed score rankings of the feature importance for the $H_2$ model are shown in Figure 5 and Table 1. The summation of all of the feature scores is 1. It is noted that the top-four features are the ER, VM, LHV, and C content, each of which contributes an essential percentage of more than 10%. According to the permutation importance analysis, the most essential input feature is the ER, which contributes 29% to the $H_2$ model. In addition, the lowest score of the $H_2$ model is the N content, which contributes 1%.

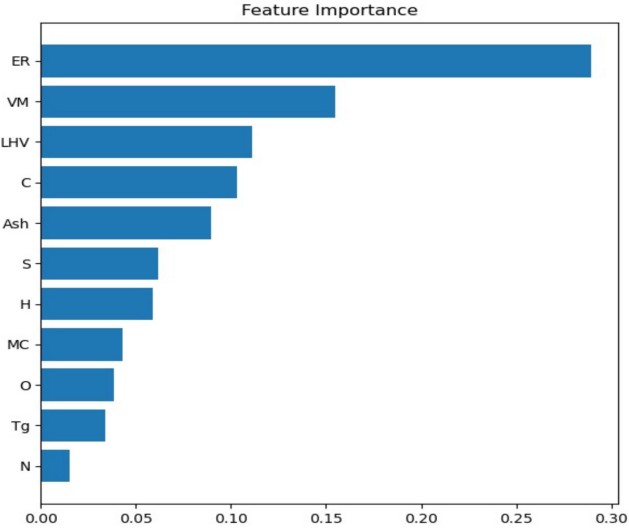

**Figure 5.** Feature importance detailed scores of the $H_2$ model.

**Table 1.** Feature importance rank of the $H_2$ model.

| Feature | Rank |
|---|---|
| ER | 1 |
| VM | 2 |
| LHV | 3 |
| C | 4 |
| Ash | 5 |
| S | 6 |
| H | 7 |
| MC | 8 |
| O | 9 |
| Tg | 10 |
| N | 11 |

*3.2. Model Performance*

In this study, a total of 11 input features are selected for the $H_2$ model, and the target is $H_2$ (% volume). The $R^2$ value is essential for evaluating how well the model fits the raw data. Upon the completion of the feature importance analysis, the number of input features is selected, according to the ranks in Figure 5, to construct the different $H_2$ models, so as to determine the best model. For the $H_2$ model, the $R^2$ value of the training data are between 0.93 and 0.97, by selecting different numbers of features. It is noted that when more numbers of the input feature are assigned, the higher the $R^2$ values. For the test data, the $R^2$ values are between 0.93 and 0.95, with a trend similar to that of the training data. When more input features are used, the higher value of $R^2$ is obtained. However, for the validation data, the $R^2$ values vary from 0.81 to 0.92, in a broad range; this represents the model performance evaluation results with different numbers of input features. Furthermore, the $R^2$ value of the entire model has the same trend as those of the training and test data. These details are shown in Figure 6. It is therefore expected that more characteristics of the data can be captured with a larger number of features.

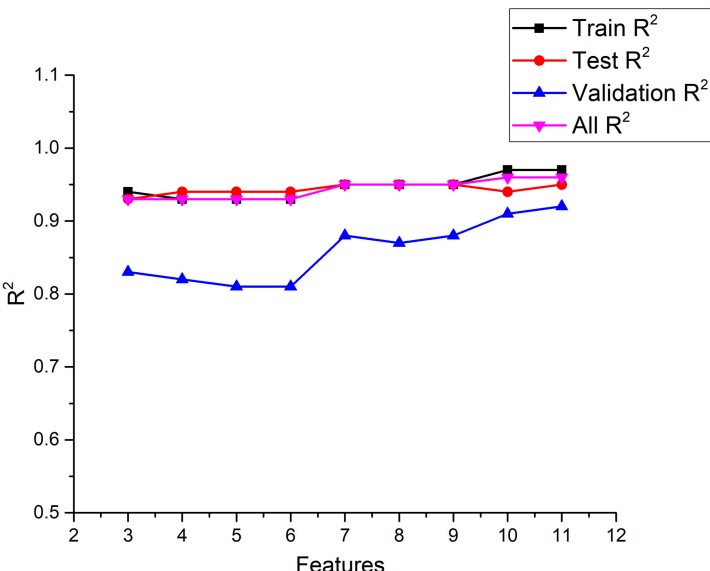

**Figure 6.** Performance $R^2$ statistics results of the $H_2$ model.

The other statistical results for the $H_2$ model performance are listed in Table 2. The RMSE and MAE values of the models using different numbers of features (top-3 and all 11 features) are listed. These two performance statistics show similar trends as the $R^2$ results.

**Table 2.** Performance RMSE and MAE statistic results of the $H_2$ model.

| Selected Features | RMSE | MAE |
|---|---|---|
| All features | 2.64 | 1.51 |
| Top 10 features | 2.74 | 1.52 |
| Top 9 features | 3.13 | 1.82 |
| Top 8 features | 3.17 | 1.86 |
| Top 7 features | 3.13 | 1.81 |
| Top 6 features | 3.79 | 2.52 |
| Top 5 features | 3.8 | 2.51 |
| Top 4 features | 3.75 | 2.45 |
| Top 3 features | 3.73 | 2.39 |

## 4. Discussion

The study by El-Shafay et al. [58] showed that the percentage of the constituent gas $H_2$ increases with the increasing $R^2$ value of the gasification temperature. In this study, the $H_2$ XGBoost regression models using different numbers of input features were built successfully. The model performance was excellent at predicting the hydrogen gas composition after gasification. It is also noted that the $H_2$ model accuracy was validated by comparisons with the results from the literature.

Table 3 shows the performance comparison between the $H_2$ model and the ANN model by Ozonoh et al. [16] using 11 input features. The results show that the $R^2$ values of the XGBoost model are higher than those of Ozonoh's ANN model, except for the test data. The reason for this is that the percentages of the test data used in these two algorithms are different. The results also show that the performance of the $H_2$ XGBoost regression model is better than that of the ANN model because the lower MSE value indicates a better model performance. However, the opposite is true for the $R^2$ value.

**Table 3.** Performance comparison between the $H_2$ model and that of Ozonoh et al. [16].

| Selected Features | $R^2$ | | | | MSE |
|---|---|---|---|---|---|
| | Train | Test | Validation | All | |
| $H_2$ model | 0.97 | 0.95 | 0.92 | 0.96 | 6.96 |
| Ozonoh et al. | 0.97 | 0.97 | 0.96 | 0.95 | 8.51 |

Furthermore, the hydrogen gas composition production was validated by comparison with the results of El-Shafay et al. [58] to verify the $H_2$ XGBoost regression model. The proximate and ultimate analyses of the sawdust pellets, based on [58], are listed in Table 4. Figure 7a,b show that the $H_2$ volume percentages vary at different ER and temperature values. The black squares represent the results of El-Shafay et al., and the red circles represent the results of the $H_2$ model. In Figure 7a, it is noted that the deviations between the two models at a temperature of 900 °C, are relatively small, but these differences are more significant at 600 and 800 °C at ER = 0.3. The trend of the increasing temperature may also result in a higher $H_2$ production linearly. Figure 7b shows that the deviations of the two models are significant at ER = 0.4. However, the trend of the $H_2$ model is observed to be similar to the results of El-Shafay et al.

**Table 4.** Proximate and ultimate analyses of the sawdust pellets [58].

| Proximate Analysis | (wt %) |
|---|---|
| Moisture | 8.8 |
| Ash | 0.58 |
| Volatile | 74.61 |
| Fixed carbon | 16.01 |
| **Ultimate Analysis** | **(wt %)** |
| Carbon | 47.37 |
| Hydrogen | 6.3 |
| Oxygen | 42 |
| Nitrogen | 0.12 |
| Sulfur | 0.0 |
| Heating value | 17.95 MJ/kg |

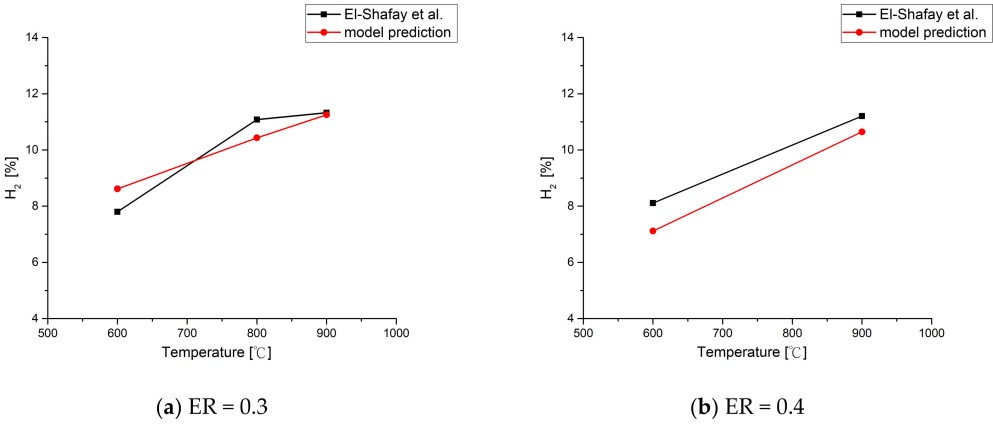

(**a**) ER = 0.3          (**b**) ER = 0.4

**Figure 7.** Comparisons of the results of the $H_2$ [%] model prediction with those of El-Shafay et al. [58] for different ER and temperature.

## 5. Conclusions

In this study, the feature importance analysis of the $H_2$ production model of biomass gasification was built successfully. The top-four features, according to importance, were determined as the equivalence ratio (ER), volatile matter (VM), lower heating value (LHV), and carbon (C) content. The model performances using different numbers of input features in training the $H_2$ model were also investigated, and the results show that selecting all 11 input features produced the best model performance, with an $R^2$ value of 0.96, because more data characteristics could be captured. For reducing the simulation cost, one may consider using the top-three input features, namely ER, VM, and LHV, in the model training while still obtaining an excellent performance ($R^2$ = 0.93). Furthermore, a comparison of the performance between the XGBoost regression model and Ozonoh's ANN model was performed, and the XGBoost regression model was observed to outperform Ozonoh's ANN model. Thus, the application of the XGBoost regression model has been validated once again. The results show that the deviations between the $H_2$ production model and the findings of El-Shafay et al. are close, especially at a temperature of 900 °C at ER = 0.4.

**Author Contributions:** Conceptualization, H.-T.W. and H.-Y.W.; methodology, H.-T.W.; software, H.-T.W.; validation, H.-T.W. and K.-C.L.; writing—original draft preparation, H.-T.W.; writing—review and editing, H.-T.W. All authors have read and agreed to the published version of the manuscript.

**Funding:** This research received no external funding.

**Data Availability Statement:** Not applicable.

**Conflicts of Interest:** The authors declare no conflict of interest.

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
