# Peer review of "Using XGBoost Regression to Analyze the Importance of Input Features Applied to an Artificial Intelligence Model for the Biomass Gasification System"

_inventions, doi:10.3390/inventions7040126_

Round 1
Reviewer 1 Report
The topic is interesting. However, major revision is needed before acceptance.
1. English need native check
2. What is the novelty of the paper
3. Line 19 R2
4. Line 39 references
5. Abstract need revision in term of correlation, statistical analysis and
6. Introduction need major revision. Firstly explain the introduction with biomass gasification system, then go for machine learning
7. Don’t break the structure by providing additional commas.
8. Explain and justify why you have use XGBOOST network.
9. Add the importance of machine learning by adding flowchart and table related to machine learning. Go through the mentioned references and add them in the manuscript.
a) Predictive modelling of sustainable lightweight foamed concrete using machine learning novel approach
b) Simulation of depth of wear of eco-friendly concrete using machine learning based computational approaches
c) Prediction of compressive strength of sustainable foam concrete using individual and ensemble machine learning approaches
d) A comparative study for the prediction of the compressive strength of self-compacting concrete modified with fly ash
e) Sugarcane bagasse ash-based engineered geopolymer mortar incorporating propylene fibers
f) Predictive modeling for sustainable high-performance concrete from industrial wastes: A comparison and optimization of models using ensemble learners
g) Prediction of compressive strength of fly ash based concrete using individual and ensemble algorithm.
10. Heading 2 total number of data is 315. Draw histogram, statistical analysis of data and covariance of the data.
11. Enhance the figure qualities
12. You have divided the data into test and train set. Then where is the validation set. Add it
13. Explain your model with the help of violin graphs
14. Compare your model with other models as well
Reviewer 2 Report
The text in the paragraph in lines 131 – 137 is the same as the text in lines 140 – 146 – the redundant text should be erased.
In the Conclusion, the abbreviations like “ER, VM, LHV, and C” should be explicitly explained – i.e., whole words followed by abbreviation in round brackets. This is recommended because some readers are focused firstly only on abstract and conclusion to decide if they should read entire article. From this reason the terms in the abstract or conclusion should be clear.
In addition, in the conclusion is stated the following: “Results show that selecting 11 input features can obtain the best model performance because more data characteristics can be captured”. This is not so surprising conclusion. The question is: It is possible to downsize the input feature number without significant loss of model accuracy? This question should be answered in conclusion - also the R2 value of model with the highest number of features should be in conclusion compared to the R2 value of model with the lowest number of features.
Round 2
Reviewer 1 Report
Overall paper quality is good. Anyhow, increase the font size of every figure as it is nor readable.